# An Investigation of All Fiber Free-Running Dual-Comb Spectroscopy

**DOI:** 10.3390/s23031103

**Published:** 2023-01-18

**Authors:** Fu Yang, Yanyu Lu, Guibin Liu, Shaowei Huang, Dijun Chen, Kang Ying, Weiao Qi, Jiaqi Zhou

**Affiliations:** 1College of Science, Donghua University, Shanghai 201620, China; 2Shanghai Institute of Optics and Fine Mechanics (SIOM), Chinese Academy of Sciences (CAS), Shanghai 201800, China; 3Hangzhou Institute for Advanced Study, University of Chinese Academy of Sciences, Hangzhou 310024, China; 4Shanghai Key Laboratory of Solid-State Laser and Application, Shanghai 201800, China; 5Center of Materials Science and Optoelectronics Engineering, University of the Chinese Academy of Sciences, Beijing 100049, China

**Keywords:** dual-comb spectroscopy, free running, spectrum measurement, fiber Bragg grating, self-correction

## Abstract

A dual-comb spectroscopy (DCS) system uses two phase-locked optical frequency combs with a slight difference in the repetition frequency. The spectrum can be sampled in the optical frequency (OF) domain and reproduces the characteristics in the radio frequency (RF) domain through asynchronous optical sampling. Therefore, the DCS system shows great advantages in achieving precision spectral measurement. During application, the question of how to reserve the mutual coherence between the two combs is the key issue affecting the application of the DCS system. This paper focuses on a software algorithm used to realize the mutual coherence of the two combs. Therefore, a pair of free-running large anomalous dispersion fiber combs, with a center wavelength of approximately 1064 nm, was used. After the signal process, the absorption spectra of multiple species were simultaneously obtained (simulated using the reflective spectra of narrow-bandwidth fiber Bragg gratings, abbreviated as FBG). The signal-to-noise ratio (SNR) could reach 13.97 dB (25) during the 100 ms sampling time. In this study, the feasibility of the system was first verified through the simulation system; then, a principal demonstration experiment was successfully executed. The whole system was connected by the optical fiber without additional phase-locking equipment, showing promise as a potential solution for the low-cost and practical application of DCS systems.

## 1. Introduction

An optical frequency comb (OFC) is generated by a phase-locked stable laser. In the time domain, it is shown as a series of ultrashort pulses with equal time intervals. In the frequency domain, the signal presents as a series of discrete and equally spaced spectral lines [1].

High-precision laser spectroscopy technology using optical combs can be divided into two categories. One category is based on the use of one comb laser as a wide band and high-coherent light source for Fourier transform spectroscopy (FTS) [2] or a dispersion spectrometer [3]. In addition, the high-quality laser pulse of the comb is coupled with the optical cavity in order to improve the detection sensitivity or enhance the spectrum and, subsequently, to carry out the high-precision spectral measurement [4,5]. Although these measurement systems based on single combs are simple, they do not fully combine the advantages of the OFC, such as its high spectral resolution, fast pulse rate and large ambiguity range.

The other category is based on the use of two combs as light sources to perform multi-heterodyne detection, and this can also be divided into two types. The first requires various scanning devices (mechanical or electronic) to ensure the alignment of the two combs in order to guarantee the two combs’ ability to coincide in the time domain and provide a coherent detection [6]. The other type renders the pulse repetition frequency (PRF) of the two combs slightly different in order to automatically realize the alignment and coherent detection of the two combs. Of these methods, the second, which is also called asynchronous optical sampling (ASOPS) dual-comb spectroscopy (DCS in short) technology [7], shows great advantages in terms of the spectral coverage and measurement speed. The basic principle of this method was first proposed by Schiller [8] in 2002. In 2004, Keilmann [9] designed experiments to verify the feasibility and effectiveness of this principle for the first time. Scientists have conducted a great deal of pioneering research on spectral measurements using this DCS system. Through the DCS system and digital correction scheme, they obtained the absorption spectra of carbon dioxide, acetylene and other gases [10,11,12,13], which proved to align with the HITRAN database. NIST fitted the absorption spectra of CO_2_ and H_2_O obtained from a high-resolution phase-locked DCS system by conducting an open-path experiment and retrieved the concentrations of 426.2 ± 3.7 ppm and 4441 ± 40 ppm, respectively [14].

A DCS system usually requires complicated devices to achieve the coherent control of the two combs. For example, Coddington et al. [15] uses two continuous-wave (CW) lasers and a high-finesse optical cavity in the lab to ensure the phase-locked and frequency stability. It appears that the complicated hardware system restricts the application scenario. However, Gar-Wing Truong et al. [16] and Sean Coburn et al. [17] conducted valuable work in order to realize the miniaturization, integration and field experiments of the DCS system. The optical components of the fieldable DCS provided in [16] fit a volume of 70 × 65 × 25 cm^3^, containing an f-2f locking, “bootstrapped” frequency referencing system. The fieldable DCS was capable of a spectral coverage of 44 THz in near-IR, with a precision of 2.8 ppm-km at 1 s, which did not deviate from the values measured using a laboratory-based system, being within 5.6 × 10^−4^. The volume of the DCS system described in [17] was 0.6 × 0.9 × 0.7 m, enabling f-2f locking, the phase locking of each comb to a narrow-bandwidth commercial CW diode laser, digitization and real-time averaging. This DCS is capable of a spectral coverage of 70 nm, operates in the field for up to 12 months, offers the ability to continuously operate and enables autonomous monitoring across multiple-square-kilometer regions, with emission rates as low as 1.6 g/min. Moreover, scientists from Spain and Ireland [18,19,20] realized two combs with mutual coherence through gain switching and optical injections. Three narrow-bandwidth commercial lasers were used. One served as the master laser, and the other two served as slave lasers. The use of a single master laser ensured a high degree of mutual coherence between the two gain-switched OFCs with different line spacings. The DCS realized by this scheme can cover regions ranging between the visible and mid-infrared, and has the advantages of a simple, compact and integrable structure, adjustable line spacing, a low cost and the potential to be applied outside of the research laboratory environment.

Based on the previous research, we can observe that in order to apply the DCS system outside of the lab, scientists have proposed different solutions, such as miniaturization, integration or the use of certain schemes, without many additional elements. These primarily relate to the hardware aspect. This paper aims to realize the mutual coherence of the DCS, focusing solely on the software aspect. Therefore, two large anomalous dispersion free-running fiber combs with slight frequency differences were used as the light source. Mutual coherence and the simultaneous measurement of multiple gas spectra were achieved through a software signal process. The work reported in this paper explored the performance of the DCS achieved through the use of free-running combs and software signal process methods. The proposed system shows promise as a potential solution for the low-cost and practical application of DCS systems.

The work was first verified through simulations and then executed in experimental demonstrations. This paper is arranged as follows: Section 1 describes the background and motivation for this research; Section 2 describes the principle and mathematical model of the DCS system; the experimental conditions, simulation results and experimental results are discussed in Section 3 and Section 4, respectively; Section 5 describes the spectral inversion and SNR analysis; the conclusion is provided in Section 6.

## 2. Principle

### 2.1. Mathematical Model

OFC, which may be simply defined as a comb, is well known for yielding spectral lines that present as a comb-like, equally spaced arrangement. The pulse sequence propagated in the time domain can be expressed mathematically as in Equation (1) [21]: (1)$$\tilde{E}(t)=\tilde{A}(t)⊗\sum_{N=-\infty }^{+\infty }\delta (t-N/f_{r})\exp(-iN\Delta \varphi )$$
where *N* is the serial number of the interferograms (IGMs), $f_{r}$ is the PRF, and ∆*φ* is the carrier envelope phase (CEP), which also decides the carrier envelope offset frequency (CEO) $f_{ceo}=\Delta \varphi \times f_{r}/2\pi$. CEO is the offset of the first comb relative to the zero frequency. $\tilde{A}\left(t\right)$ is the pulse function, which can be expressed as Equation (2): (2)$$\tilde{A}(t)=A_{0}(t)\exp[i(f_{c}t+\varphi_{0})]$$
where $A_{0}\left(t\right)$ represents the time domain envelope, $f_{c}$ represents the carrier frequency of the comb, and $\varphi_{0}$ is the initial phase. Thus, we can establish the mathematical model of an OFC using Equation (3): (3)$$\tilde{E}(t)=\sum_{N=-\infty }^{+\infty }A_{0}(t-NT_{r})\exp\left\{i\left[f_{c}(t-NT_{r}+\varphi_{0})\right]\right\}\exp(-iN\Delta \varphi )$$
where *T_r_* = 1/f_r_ is the time period. The DCS system uses two OFCs with slightly different PRFs as light sources, serving as the signal comb (SIG) and local oscillator comb (LO), respectively. Assuming that the SIG and LO combs have the same time domain envelope, after multi-heterodyne spectroscopy, the spectrum information is transferred from the OF domain to the RF domain through the detector. A series of RF combs are obtained, and Figure 1 [22] demonstrates the down-converting process in both the time and frequency domains. The optical sampling is realized by the PRF difference between the two combs, which is clearly shown in the time domain in Figure 1a,b. In this process, one comb appears to pass through the other comb. The multi-heterodyne interference process is clearly shown in the frequency domain in Figure 1c,d. According to [22], the one-to-one correspondence between the OF and RF combs can be guaranteed using a band-pass filter (BPF) and a low-pass filter (LPF) to prevent aliasing and confusion, respectively.

The beat signal of the two combs can be expressed as Equation (4) in the RF domain: (4)$$\begin{array}{l} U(t)={\tilde{E}}_{LO}^{*}(t)⊗{\tilde{E}}_{SIG}^{}(t)\\ \propto \sum_{N=-\infty }^{+\infty }A_{0}{}^{2}(t-NT_{r}{}^{RF})\exp\left\{i\left[f_{c}{}^{RF}(t-NT_{r}{}^{RF}+\varphi_{0}{}^{RF})\right]\right\}\exp(-iN\Delta \varphi_{0}{}^{RF}) \end{array}$$
where $T_{r}^{RF}=1/\Delta f_{r}$ is the period of the IGM in the RF domain, Δf*_r_* is the frequency difference of the two combs, $f_{c}^{RF}$ is the carrier frequency in the RF domain, $\varphi_{0}^{RF}$ is the initial phase in the RF, and $\Delta \varphi_{0}{}^{RF}$ is the carrier envelope phase in the RF. Supposing the comb frequencies of SIG and LO are $f_{1m}$ and $f_{2n}$ respectively, they can be expressed as Equation (5). The RF combs can be expressed as Equation (6) [22]: (5)$$f_{1m}=mf_{r1}+f_{ceo1};\;f_{2n}=nf_{r2}+f_{ceo2}$$
(6)$$f^{RF}=f_{1m}-f_{2n}=f_{ceo1}-f_{ceo2}+mf_{r1}-nf_{r2}=f_{ceo}^{RF}+m\Delta f_{r};\;$$
where $f_{ceo}^{RF}$ is the carrier envelope frequency in the RF. Equation (6) shows that the RF frequency combs has a similar format to the OF frequency combs. Supposing that $\tau_{0}^{RF}$ is the initial phase center position of the IGMs, it can be expressed as Equation (7): (7)$$\tau_{0}^{RF}=\frac{f_{r1}\tau_{1}-f_{r2}\tau_{2}}{\Delta f_{r}}$$
where $\tau_{1}^{}$ and $\tau_{2}^{}$ are the initial phase center positions of the SIG and LO combs. The initial phase in the RF domain $\varphi_{0}^{RF}$ and carrier envelope phase in the RF domain $\Delta \varphi_{0}{}^{RF}$ can be expressed as Equations (8) and (9): (8)$$\varphi_{0}^{RF}=\varphi_{SIG}-\varphi_{LO}+2\pi \left[f_{ceo1}\tau_{1}-f_{ceo2}\tau_{2}+(m-n)\cdot f_{r2}\tau_{2}-f_{ceo}^{RF}\tau_{0}^{RF}\right]$$
(9)$$\Delta \varphi_{0}^{RF}=\frac{2\pi \Delta f_{ceo}}{\Delta f_{r}}$$
where $\Delta f_{ceo}=f_{ceo2}-f_{ceo1}$ is the difference of the two combs’ CEO.

For the free-running fiber laser, the PRFs of the combs and the shift between the two combs are not locked. As a result, the PRFs and CEO frequencies have random drifts in both the time and frequency domains. Considering the effect of these jitters, Equation (4) can be expressed as Equation (10): (10)$$\begin{array}{l} U_{jitter}(t)\propto \sum_{N=-\infty }^{+\infty }A_{0}{}^{2}\left[t-NT_{r}{}^{RF}+\delta T_{r}{}^{RF}(N)\right]\\ \times \exp\left\{i\left[f_{c}{}^{RF}+\delta f_{c}{}^{RF}(N)\right]\times \left[t-NT_{r}{}^{RF}+\delta T_{r}{}^{RF}(N)\right]\right\}\\ \times \exp\left\{-i\left[N\Delta \varphi_{0}{}^{RF}+\varphi_{0}{}^{RF}+\delta \varphi_{0}{}^{RF}(N)\right]\right\} \end{array}$$
where $\delta T_{r}^{RF}(N),\;\delta f_{c}^{RF}(N),\;\delta \varphi_{0}^{RF}(N)$ are the time jitter, carrier frequency jitter and phase jitter of the *N*th IGMs in the RF domain, respectively. Referring to [22,23], the three jitters can be expressed as Equations (11)–(13): (11)$$\delta T_{r}^{RF}(N)=\frac{1}{\Delta f_{r}+\delta f_{r}(N)}-\frac{1}{\Delta f_{r}}$$
where $\delta f_{r}\left(N\right)=\delta f_{r2}\left(N\right)-\delta f_{r1}\left(N\right)$ represents the shift of the PRF difference between the two combs. This is the reason for the fluctuations in the IGM period $\delta T_{r}^{RF}N$, which is represented as the regular comb arrangement and shape deformation in the frequency domain.
(12)$$\begin{array}{l} \delta f_{c}^{RF}(N)=\Delta f_{ceo}+\delta f_{ceo}^{}(N)\\ +\frac{\Delta f_{r}+\delta f_{r}(N)}{f_{r}+\delta f_{r}(N)}\times \left[f_{c}+\delta f_{c}(N)-f_{ceo1}-\delta f_{ceo1}(N)\right]-f_{c}^{RF} \end{array}$$
where $\delta f_{ceo}\left(N\right)=\delta f_{ceo2}\left(N\right)-\delta f_{ceo1}\left(N\right)$ represents the shift of the CEO difference between the two combs. It is observed that the carrier frequency jitter $\delta f_{c}^{RF}\left(N\right)$ includes the drift of both $f_{ceo}$ and $f_{r}$. It is characterized as the envelope changes in the time domain: 

The phase jitter $\delta \varphi_{0}^{RF}N$ is as follows: (13)$$\delta \varphi_{0}^{RF}(N)=\frac{2\pi \left[\Delta f_{ceo}+\delta f_{ceo}(N)\right]}{\Delta f_{r}+\delta f_{r}(N)}-\Delta \varphi_{0}^{RF}$$

The jitters mentioned above are the main target of our digital self-correction algorithm. The simulation of the DCS in this study was based on Equations (4)–(13). The equations clearly show that the RF interference diagram is the sum of the carrier envelope periodic signals. Various jitters in the OF domain are also reproduced in the RF domain, and the mapping scale factor, in theory, is $f_{r}/\Delta f_{r}$.

### 2.2. DCS Principle

There are two common types of DCS systems: the symmetric system and the asymmetric system [7]. The SIG comb of the asymmetric system traverses the sample before coherent with the LO comb. The reference signal and target signal are combined after absorption and then collected by a detector. The reference signal and target signal are distinguished by time. The advantage of the asymmetric system is that the amplitude spectrum and phase spectrum can be obtained simultaneously, and the influences of the different detectors can be eliminated. However, the phase spectrum may contain excessive noise if the two combs are totally free-running, which will lead to significant distortion.

Figure 2 depicts the symmetric spectrum measurement system. The LO and SIG combs are first coherent in coupler 1 and then pass through the isolator and BPF. The coherent signal is then split into two paths by coupler 2. One path, called the signal path, passes through the sample cell, the low-noise photodetector 1 (PD1) and the LPF1, and it is then converted into a digital signal. The other path, called the reference path, directly passes through the PD2 and LPF2 before being converted into a digital signal. PD1 and PD2 have the same performance specifications, in theory, and the same situation applies to the LPF1 and LPF2. The two paths’ signals are sampled by the same analog-to-digital converter (ADC) and then processed by the computer.

According to [24], the signal of the reference path without passing the sample can be expressed as the convolution of the two optical comb signals, as shown in Equation (14): (14)$$V_{ref}(t)=E_{LO}^{*}(t)⊗E_{SIG}^{}(t)$$
where $E_{LO}{}^{*}\left(t\right)$ ⊗ $E_{SIG}\left(t\right)$ are expressed in Equation (4). The signal of the path through the sample can be expressed as Equation (15): (15)$$V_{sig}(t)=E_{LO}^{*}(t)⊗E_{SIG}^{}(t)⊗H(t)$$
where *H*(*t*) is the sample response function in the time domain, and the sample frequency response function *H*(*ν*) is separated in the frequency domain by Equation (16): (16)$$\tilde{H}(v)=\frac{V_{sig}(v)}{V_{ref}(v)}={\tilde{H}}_{0}(v)+\sigma_{H}$$
where ${\tilde{H}}_{0}\left(v\right)$ is the frequency response of the real sample and $\sigma_{H}$ is the unavoidable additional noise, which affects the signal-to-noise ratio (SNR) and the accuracy of the spectral measurement.

### 2.3. SNR

SNR is a key parameter used to evaluate the system performance. In the DCS experimental system, the noise can be divided into two categories: additive noise and multiplicative noise. Additive noise appears during the signal detection stage, including detector noise, shot noise, laser relative intensity noise (RIN) and dynamic range noise. The multiplicative noise is the noise created in the sampling process caused by the residual carrier phase noise between the two combs.

Supposing the $SNR_{add}$ and $SNR_{mult}$ are the SNR limited by the additive noise and multiplicative noise of the DCS system. Referring to [24], they can be expressed as Equations (17) and (18):
(17)$$SNR_{add}=\frac{1}{\sigma_{H,add}}=0.8\frac{N_{d}\sqrt{T}}{M\sqrt{\varepsilon }}\sqrt{\frac{F}{2RIN+8D^{-2}f_{r}^{-1}}}$$
(18)$$SNR_{mult}=\frac{1}{\sigma_{H,mult}}=\frac{1}{3}\sqrt{\frac{TN_{d}f_{r}}{\Delta v}}\sigma_{\varphi ,fast}^{-1}$$
where $\sigma_{H,\mathrm{add}}$ and $\sigma_{H,mult}$ refer to the additive noise and multiplicative noise; *M* is the number of spectral elements; ε is the duty cycle; RIN is the measured laser relative intensity noise; and D is the dynamic range of the detector. Δv is the spectral bandwidth of the OFC. *F* is the low sequence acquisition stage and *N_d_* is the low parallel acquisition stage. *T* is the sampling time. $\sigma_{\varphi ,fast}^{}$ is the inter-pulse variance in the carrier phase white noise. The final SNR is determined by *SNR_add_* and *SNR_mult_* together.

### 2.4. Signal Process

Typically, it is difficult to obtain a sample absorption spectrum with only one IGM, and multiple measurements must be accumulated to improve the SNR. As stated in [25], the spectrum signals are accumulated over 93 ms (200 IGMs). However, in the free-running DCS system, the IGMs are different from each other due to various noises. The SNR cannot be improved if the results are directly accumulated without correction. For the noises mentioned in Section 2.1, the main steps of the signal process algorithm are as follows: 

(1) Noise reduction in both the time domain and frequency domain. For one period of IGMs, valuable information exists for a very short time compared to the entire period. Similarly, the valuable signal exists within a certain range in the frequency domain. Signals outside of the usable range can be eliminated.

(2) Self-correction algorithm. Select one IGM closest to the average value (including the RF period length and RF carrier frequency) as a reference and then perform the digital self-correction scheme for the other IGMs [22,23,26]. The procession of the digital correction method can be described in the following steps: 

Firstly, the time jitter $\delta T_{r}^{RF}N$ can be compensated by interpolation and resampling, which is shown in Equation (19): (19)$$U_{step1}(t)=U_{jitter}[t-\delta T_{r}^{RF}(N)]$$

Secondly, perform the Hilbert transform on the $U_{step1}\left(t\right)$ to generate an analytical signal $S_{step1}\left(t\right)$ and compensate the carrier frequency jitter $\delta f_{c}^{RF}N,$ as described in Equation (20):(20)$$U_{step2}(t)=real\left\{S_{step1}(t)\times \exp[-2\pi j\delta f_{c}^{RF}(N)t]\right\}$$

Finally, the carrier envelope phase of $U_{step2}\left(t\right)$ is calculated by the FFT, then phase jitter $\delta \varphi_{0}^{RF}N$ can be compensated as Equation (21): (21)$$U_{step3}(t)=real\left\{S_{step2}(t)\times \exp[-2\pi j\delta \varphi_{0}^{RF}(N)t]\right\}$$

After three steps of digital error correction, phase-aligned IGMs without jitter can be achieved.

(3) Acquire the RF transmission spectrum by coherent superposition and calculate the SNR. Then, map the RF transmission spectrum to the OF and obtain the transmission spectra of multiple gases (simulated using the reflective spectrum of narrow-bandwidth FBG) at the same time.

The method of mapping from the RF domain to the OF domain is based on the conversion factors and corresponding relation between the characteristic positions of the optical spectrum and RF domain spectrum. The conversion formula is shown in Equation (22): (22)$$v_{OF}=v_{OFS}\pm (v_{RF}-v_{RFS})\frac{f_{r}}{\Delta f_{r}}$$
where $\nu_{OFS}$ and $\nu_{RFS}$, respectively, represent the characteristic points of the frequency in the OF domain and RF domain, while $\nu_{RF}$ and $\nu_{OF}$ are the frequency in the RF domain and OF domain, respectively. $f_{r}/\Delta f_{r}$ is the conversion factor. It should be noted that the RF signal is the beat signal of the two free-running OFCs. Thus, the sampling spectrum in the RF domain may be a mirror reversal state in contrast with the original spectrum in the OF domain. Regarding the “±” in Equation (22), “−” is applied when the mirror reversal state appears. Otherwise, “+” is applied.

## 3. Simulation

Before the experiment, the simulation system was constructed to evaluate the feasibility by referring to the parameters of the real experimental system and multiple noise sources [27,28], as shown in Table 1.

The optical spectrum of the signal path and the reference path (without FBG) were sampled using a spectrometer with a resolution of 0.002 nm, which is shown in Figure 3.

Referring to [22], in order to avoid confusion, the spectral bandwidth in the RF should be less than *f_r_*/2, which equals approximately 13 MHz. Therefore, the corresponding maximum OF spectrum width must be approximately 4.6 nm to avoid aliasing. According to Figure 3, the 3 dB bandwidth of the laser source spectrum of the system is approximately 0.7 nm. Therefore, the BPF is not necessary in this system setup.

Through the investigation of the HITRAN 2020 database, the absorption line-strength of gases is really low in the range of the lasers [29]. Referring to [30], the bandwidth of a given absorption line of CO_2_ is approximately 4.5 GHz. The 3 dB bandwidths of the reflective spectra of the two narrowband FBGs are 9.27 G and 11.3 G, respectively. This means that the 3 dB bandwidth of the FBG reflective spectrum and the real gas absorption linewidth are of the same order of magnitude. Therefore, the two reflective spectra of the narrowband FBGs can be used to simulate the absorption effect of the measured gas.

According to the DCS principle, the RF spectrum should be proportional to the OF spectrum. In the simulation system, referring to Table 1 and Equation (10), the spectra with similar double-Gauss peak shapes are constructed, as shown in Figure 4.

In Figure 4, the RF spectrum is based on the Fast Fourier Transform (FFT) of 28 IGMs generated by the simulation system. The sample period is 100 ms, which is also the experimental system’s maximum sampling length. The interference spectrum is relatively noisy due to the presence of numerous jitters, but the two simulated absorption lines are still distinct. From the inset figure, it is observed that the comb structures are buried by the noise. The simulation results are processed by the signal process algorithm described in Section 2.4. Figure 5 and Figure 6 show the processing results of the reference path and signal path signals, respectively.

It can be seen from Figure 5b and Figure 6b that the raw IGMs of the symmetrical DCS system basically maintain the original characteristic of the OF domain. However, observing the magnified part, we can see that the RF comb structures are buried by the noise. After the signal process algorithm, the background noise in the time domain is greatly reduced, as shown in Figure 5c and Figure 6c, and the comb structure is effectively restored, as shown in Figure 5d and Figure 6d.

The corrected results can be taken as the basis for the optical spectrum inversion. Firstly, the reference and signal paths are normalized. Secondly, the position of the RF comb teeth is calibrated according to the SIG comb. Finally, the samples’ absorption spectra are obtained by comparing the SIG and REF RF spectra. The normalized comparison of the signal process results and the absorption spectra in the RF domain are shown in Figure 7.

From Figure 7a, we can observe that the samples’ absorption positions are clearly within the range of 7–9 MHz. Therefore, the absorption spectra of the samples are displayed within this range, and the frequency resolution of the single comb component after correction is 10 Hz, as shown in Figure 7b. In order to show the improvement of the corrected signal, the magnified raw signal is also offered in Figure 7b. The transmittance spectrum can be obtained by comparing the RF components between the SIG path and REF path, as shown in Figure 8. Compared with the original optical spectrum in Figure 3, it can be found that the troughs of A and B correspond to FBG1 and FBG2, respectively. C is the corresponding point of $\nu_{OFS}$ in RF domain, which is the peak position do not affected by FBG.

As shown in Figure 8, the minimum transmittance values of A and B are 0.286 and 0.296, respectively, which appear as the absorption caused by the samples. To verify the result, the next step is the mapping of the samples’ spectra from the RF to the OF. As shown in Equation (22), the key mapping points are the decisions of the characteristic points in the RF and OF spectra, and the ideal scaling factor *k* = $f_{r}/\Delta f_{r}$. The point C value at 6.002 MHz in Figure 7a and the peak unpolluted by absorption in Figure 3 are selected as the characteristic corresponding points, $\nu_{OFS}=281.62\;\mathrm{THz}$. Due to the random jitters of $f_{r}$ and $\Delta f_{r}$, the scaling factor is replaced by the ratio of the spacing between the two typical absorption peaks, as shown in Figure 3 and Figure 8. The points A and B in Figure 8, with values of 8.59 MHz and 7.97 MHz in the RF spectrum, correspond to the center positions of the reflective spectra of FBG1 and FBG2 in Figure 3, with values of 1064.5489 nm and 1064.797 nm. The inverted OF spectra of the samples’ absorption lines are shown in Figure 9.

The values of A and B are 0.714 and 0.704, respectively, which means the absorptivity values are 0.714 and 0.704. Compared with the values shown in Table 1, the errors of the two FBGs’ designed reflectivity are 2.2% and 0.2%. The errors are probably caused by the combined effects of the residual noise and normalization process.

The simulation results demonstrate that the simulation system can obtain effective absorption information regarding the OF domain through the signal process algorithm, which also proves the effectiveness of the FBG-simulated gas absorption lines and DCS spectral detection.

## 4. Experiment

The all-fiber free-running DCS experiment system schematic is shown in Figure 10. The center wavelength of the laser source used in this experiment was approximately 1064 nm, with a 3 dB bandwidth of approximately 0.7 nm. The reflective spectrum of the two narrow-bandwidth FBGs were used to simulate the effect of the gas absorption lines in the experiment.

Two free-running OFCs were generated by a large anomalous dispersion ytterbium-doped fiber laser [31]. The pulse repetition rate was measured to be at approximately 26.4 MHz, which provided a pulse energy of 3.5 nJ. The PRFs of the two OFCs have a slight difference, of approximately 281 Hz. Two frequency meters are used to monitor the PRFs so that the frequency jitter can be understood. The measured phase noise of the comb (tested by Symmetricom 5125A) in the frequency range between 0.1 Hz and 1 MHz is shown in Figure 11. This was compared with another comb based on a dispersion-managed fiber laser [32].

As the two combs used in this experiment are based on large anomalous dispersion fiber lasers, the integrated phase noise was approximately 47 times higher than that of the dispersion-managed fiber combs [32]. The adoption of the large phase noise laser was intended to prove the advantages of the signal process algorithms. Regarding the detectors, the PR-125M low-noise detectors produced by Beijing Conquer Optoelectronic Technology were selected. An oscilloscope was used as the multi-channel ADC to convert the analog signals sampled from both the reference path and the signal path into digital signals. All of the signal processing was conducted on the computer.

The sampling time in this experiment was 100 ms, which includes 28 IGMs that were sampled at one time by the ADC. The spectrum stayed relatively stable within the sampling time. The processing results of the reference path and signal path of the experiment are shown in Figure 12 and Figure 13, respectively.

Several points can be made based on Figure 12 and Figure 13. Firstly, it can be seen from Figure 12b and Figure 13b that the RF spectra of the free-running symmetrical DCS system maintain the basic characteristics of the OF domain, but they are distorted by strong noises. From Figure 12e and Figure 13e, we find that the RF comb structures are influenced by various noises, and the absorption line cannot easily be obtained through these noisy RF spectra. After the signal process, the background noise in the time domain is reduced greatly, as shown in Figure 12c and Figure 13c. Moreover, the comb structure can be restored, as shown in Figure 12f and Figure 13f.

Secondly, comparing Figure 12a with Figure 13a, the IGMs of the reference and signal paths show distinct characteristics in the time domain. They are sampled by detectors produced by the same company but based on a different model. The influences of various detector models can be reduced by the noise reduction process in the frequency domain and the normalization in the spectrum retrieval phase.

Thirdly, the absorption lines’ positions are also distorted through the noises, differing from the obvious absorption positions shown in Figure 3 and Figure 4. This is possibly due to additional noises that we neglected during the simulation process. However, the absorption spectra of the samples are determined by the ratio of the signal path and reference path. The same deformation tendency would not clearly affect the final samples’ absorption spectra.

Finally, besides the two valleys at the frequencies of the expected absorption peaks, some ripples appear in the envelop of the signal path. The situation is the same for both the simulation and experiment results. This is probably due to the fact that some information on the samples is discarded during the noise reduction process. We must distinguish between the influences of these small ripples and the sample absorption line in the process. Normalization is also required before calculating the transmission spectrum.

Comparing Figure 12d and Figure 13d with Figure 3, it is found that the sample information is concentrated in the range of 7.5–9 MHz. Thus, the normalization process is also primarily performed within this range. The normalized self-correction experiment results for the RF domain are shown in Figure 14. The absorption spectra of the samples are also displayed within this range, as shown in Figure 15.

Comparing Figure 14a with Figure 3, it is found that the troughs of D and E correspond to FBG1 and FBG2, respectively. F is the corresponding point of $\nu_{OFS}$ in RF domain, which is the peak position do not affected by FBG. The absorption peak D is incomplete. This is probably due to the fact that the additional ripples mentioned above are close to the absorption peak. Comparing Figure 14b with Figure 7b, we can also clearly find that the resolutions of the corrected spectra are 10 Hz, which shows the effectiveness of the software signal process on the improvement of the SNR and distinguishing the comb structure.

As shown in Figure 15, the minimum transmittance values of E and D are 0.28 and 0.41, respectively, which appear as the absorption caused by the samples. Then, we can map the samples’ spectra from the RF to the OF with Equation (22). The point F value at 6.823 MHz in Figure 14a and the peak unpolluted by absorption in Figure 3 are selected as the characteristic corresponding points, $\nu_{OFS}=281.62\;\mathrm{THz}$. The points E and D in Figure 14a are located at 8.03 MHz and 8.71 MHz in the RF spectrum, corresponding to the FBG1 and FBG2 in Figure 3, with values of 1064.5489 nm and 1064.797 nm. The inverted OF spectra of the samples’ absorption lines are shown in Figure 16.

From Figure 16, we can find that the minimum transmissivity corresponding to FBG1 is 0.41, while that corresponding to FBG2 is 0.28. Therefore, the errors compared to the designed FBG reflectivity are 14.6% and 1.4%, respectively. In addition to the noise reduction process and the detector influence, the normalization process in the experiment probably contributes to the errors. As the normalization process is based on the maximum value of the FBG2 area, the FBG1 bears a larger error.

## 5. SNR Analysis

After obtaining the correction results, the SNR can be calculated by referring to Section 2.3. Table 2 provides the parameters used for calculating the theorical SNR of the experiment. Substituting these parameters in Equations (17) and (18), the additive noise $\sigma_{H,add}$ = 0.0224, and the multiplicative noise $\sigma_{H,mult}$ = 0.0158. Therefore, the experiment SNR, in theory, is 14.2 dB (26.2).

The enlarged comb structure obtained after the signal process is shown in Figure 17.

From Figure 17, we can see that the averaged SNRs of the comb are approximately 8.2 dB (6.67) and 13.97 dB (25), around the absorption peak and the peak of the whole spectrum, respectively. The SNR is calculated by the ratio of the average peak and the average noise of the vicinity area. The SNR around the peak position of the whole spectrum is consistent with the theory calculation.

The SNR at the various sampling times is shown in Figure 18. The number of IGMs N is proportional to the sampling time. It can be seen that the SNR increases with the length of the sampling time. The fitting curve, which is marked by ○ in Figure 18, is proportional to √N, agrees with the theorical result calculated by equations in Section 2.3, which is marked by * in Figure 18. This means that the SNR can be improved further by increasing the sampling time.

## 6. Conclusions

In this study, we intended to realize the mutual coherence of the DCS solely on the software aspect. Therefore, two large anomalous dispersion free-running fiber combs with slight frequency differences were used as the light source to realize the simultaneous, effective detection of the characteristic absorption spectral lines of various gases (simulated by the reflective spectra of two narrow-bandwidth FBGs). Noise reduction and self-correction algorithms were executed before obtaining the samples’ absorption spectra in the RF domain. Taking both the RF and the OF characteristic positions and scaling factors into consideration, inversion from the RF domain to the OF domain was successfully carried out. Due to the fluctuation in the frequency jitter, in theory, the scaling factor $f_{r}/\Delta f_{r}$ was replaced by the ratio of the spacing of the characteristic positions of the OF and RF domain.

Both simulation and demonstration experiments were conducted in our research. In order to initially demonstrate its feasibility, the simulation system’s parameters were based on the experiment parameters. In the experiment, two large-anomalous-dispersion fiber combs were used as lasers. Both the RF and OF spectra were obtained, with a significantly improved SNR and comb structure. The SNR after the signal process reached 13.97 dB (25) when the laser center band was approximately 1064 nm, the repetition frequency of the optical comb was approximately 26 MHz, the repetition frequency difference was approximately 281 Hz, the integrated phase noise of the comb reached 0.47 rad, ranging between 0.1 Hz and 1 MHz, and the single sampling time was 100 ms. The integrated phase noise of the two combs was approximately 47 times higher than that of the dispersion-managed fiber combs, which highlights the advantage of the signal process algorithm reported in this work. The experimental SNR agreed well with the theory-based SNR, and we found that it can be improved further by increasing the sampling time. The resolution of the system is enough to support the more accurate detection of gas absorption lines.

Combining the results of the simulation system and this experiment, the feasibility of the free-running DCS system is verified. Compared with the complex phase-locking system, the all-fiber free-running DCS system used in this experiment can obtain multiple samples’ absorption spectra (simulated by FBG) at the same time through the software signal process. The whole system is completely constructed from commercially mature devices operating at 1064 nm, which could easily be extended to other wavelengths. The work reported in this paper shows promise as a potential solution for the low-cost and practical application of DCS systems.

## Figures and Tables

**Figure 1 sensors-23-01103-f001:**
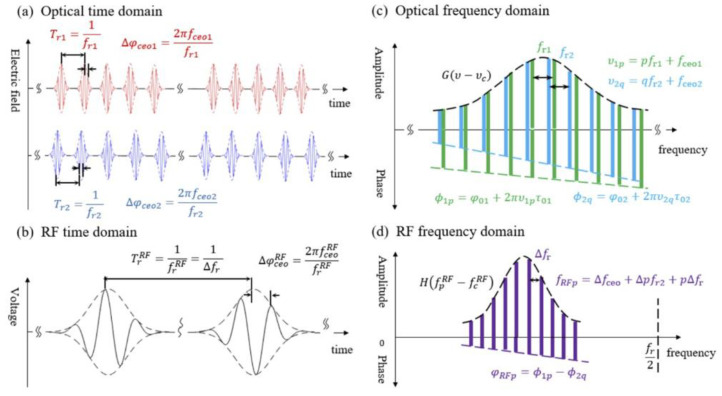
Principle of dual-comb interference [22]. (**a**) Time domain of the two combs’ periodic sequence; (**b**) time domain of the interference envelope diagram in the RF; (**c**) OF domains of the LO and SIG combs; (**d**) RF spectrum of SIG.

**Figure 2 sensors-23-01103-f002:**
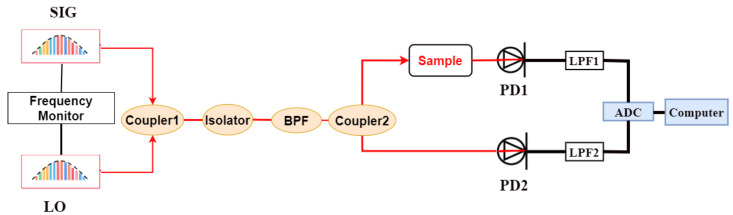
Diagram of the DCS system.

**Figure 3 sensors-23-01103-f003:**
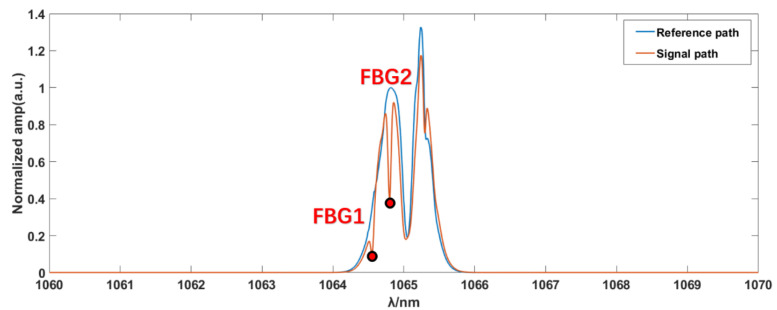
Optical spectra of the reference path (blue) and signal path (red).

**Figure 4 sensors-23-01103-f004:**
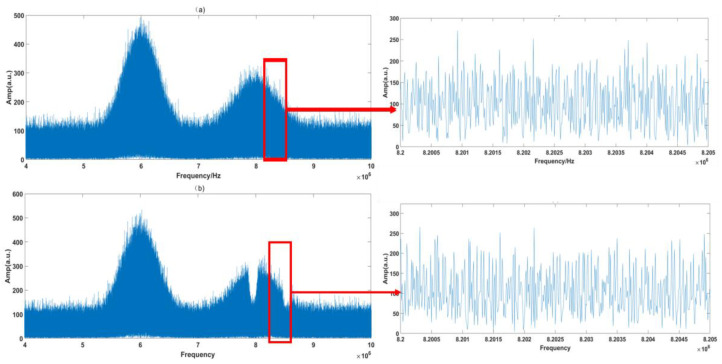
Simulation of the DCS system’s RF spectra with jitters. (**a**) Reference path spectrum, (**b**) signal path spectrum.

**Figure 5 sensors-23-01103-f005:**
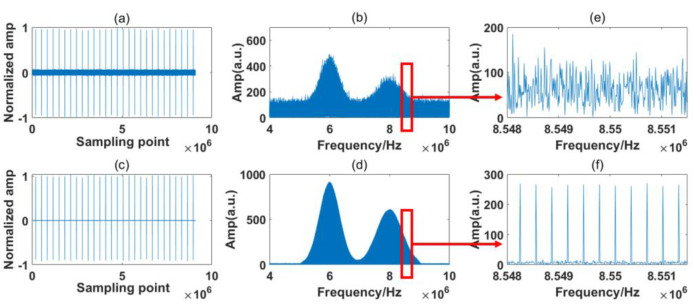
Simulation of the reference path signal. (**a**) Raw time domain signal; (**b**) raw frequency domain signal; (**c**) corrected time domain signal; (**d**) corrected frequency domain signal; (**e**) Magnified view of red rectangle shown in (**b**); (**f**) Magnified view of red rectangle shown in (**d**).

**Figure 6 sensors-23-01103-f006:**
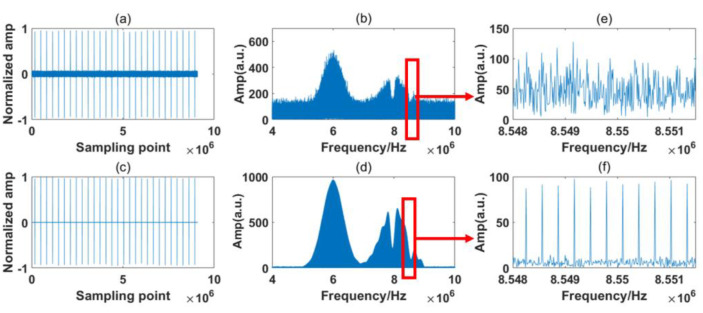
Simulation of the signal path signal. (**a**) Raw time domain signal; (**b**) raw frequency domain signal; (**c**) corrected time domain signal; (**d**) corrected frequency domain signal; (**e**) Magnified view of red rectangle shown in (**b**); (**f**) Magnified view of red rectangle shown in (**d**).

**Figure 7 sensors-23-01103-f007:**
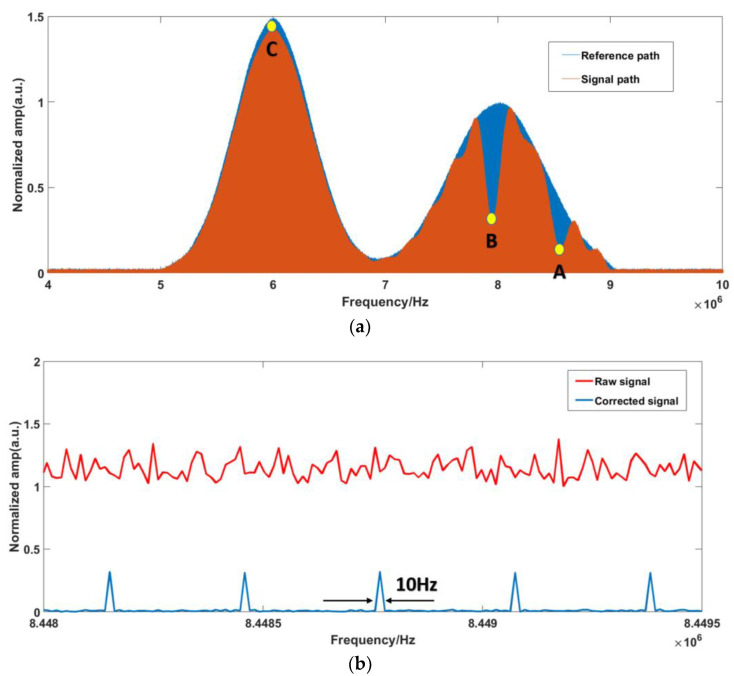
(**a**) Normalized RF spectra of simulation signal; (**b**) Magnified view of the components.

**Figure 8 sensors-23-01103-f008:**
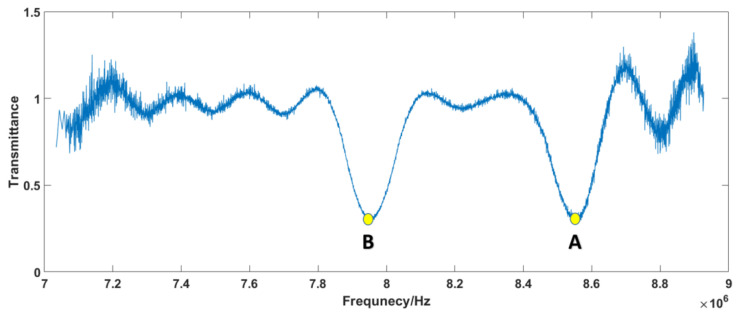
Transmittance spectrum of the simulation signal.

**Figure 9 sensors-23-01103-f009:**
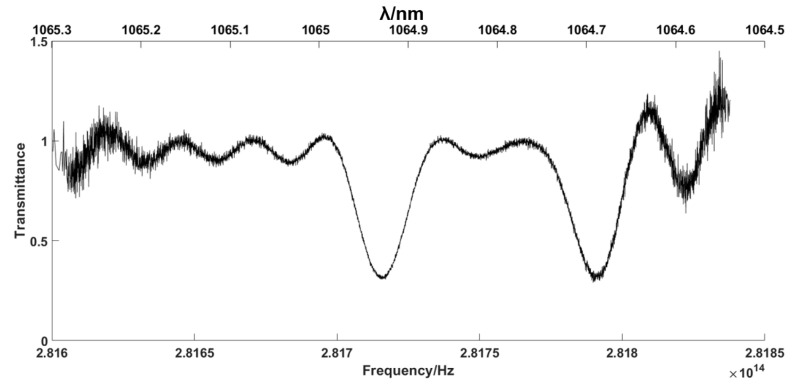
Simulation inversion from the RF domain to the OF domain.

**Figure 10 sensors-23-01103-f010:**
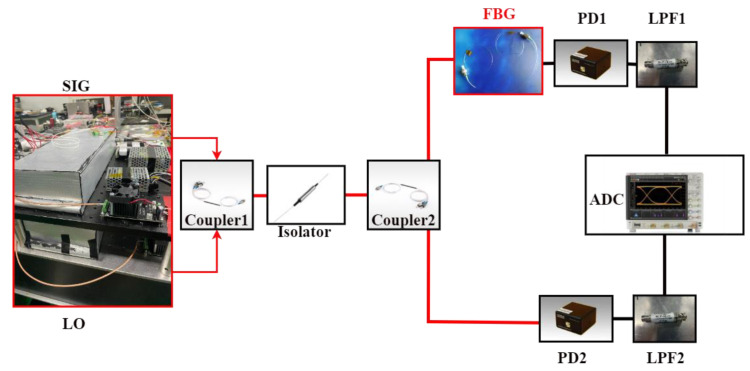
Real experiment system schematic.

**Figure 11 sensors-23-01103-f011:**
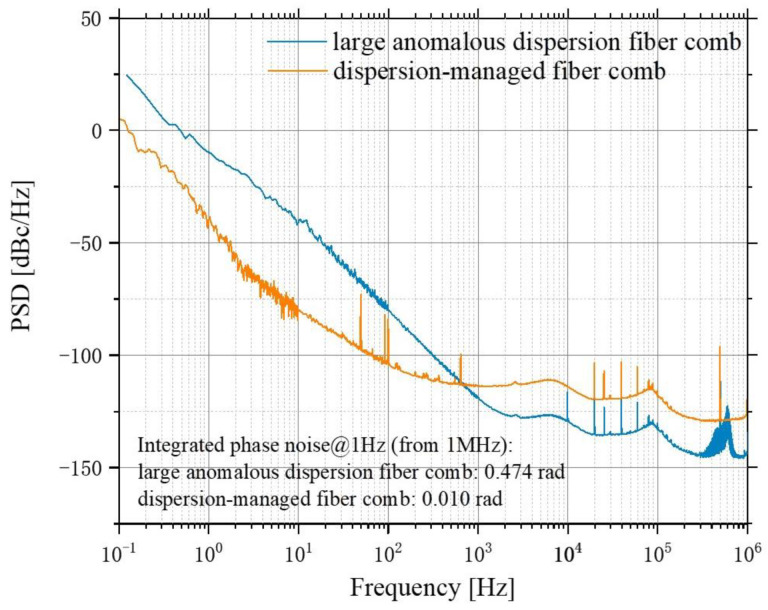
Phase noise measurement of the large anomalous dispersion fiber comb used in this work and another typical dispersion-managed fiber comb [32].

**Figure 12 sensors-23-01103-f012:**
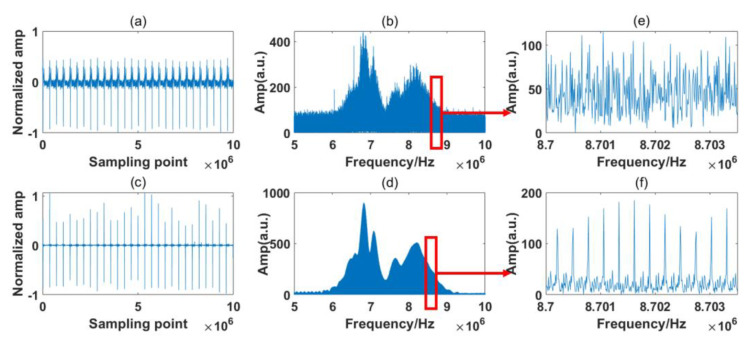
Experiment of the reference path signal. (**a**) Raw time domain signal; (**b**) raw frequency domain signal; (**c**) corrected time domain signal; (**d**) corrected frequency domain signal; (**e**) Magnified view of red rectangle shown in (**b**); (**f**) Magnified view of red rectangle shown in (**d**).

**Figure 13 sensors-23-01103-f013:**
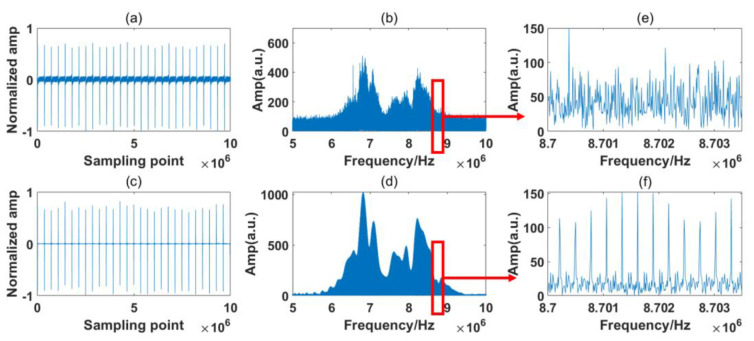
Experiment of the signal path signal. (**a**) Raw time domain signal; (**b**) raw frequency domain signal; (**c**) corrected time domain signal; (**d**) corrected frequency domain signal; (**e**) Magnified view of red rectangle shown in (**b**); (**f**) Magnified view of red rectangle shown in (**d**).

**Figure 14 sensors-23-01103-f014:**
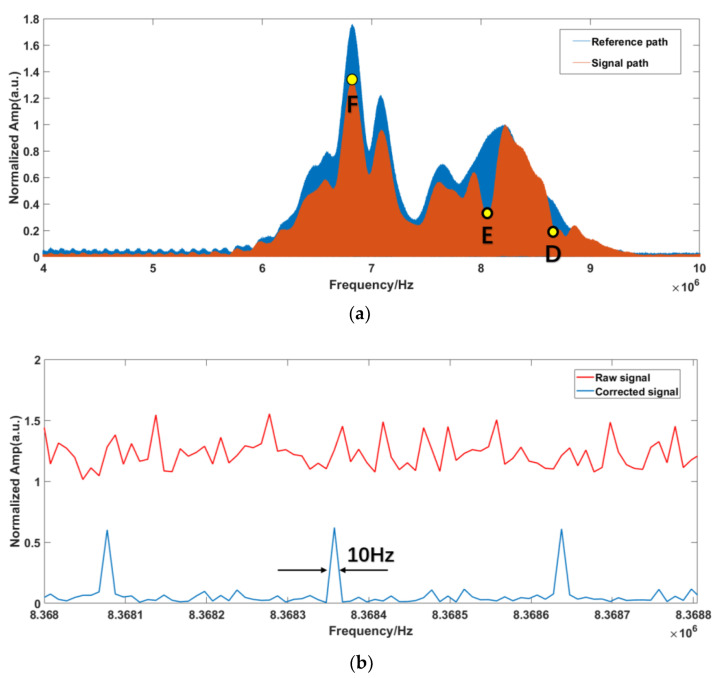
(**a**) Normalized spectrum of the experiment signal. (**b**) Magnified view of the components.

**Figure 15 sensors-23-01103-f015:**
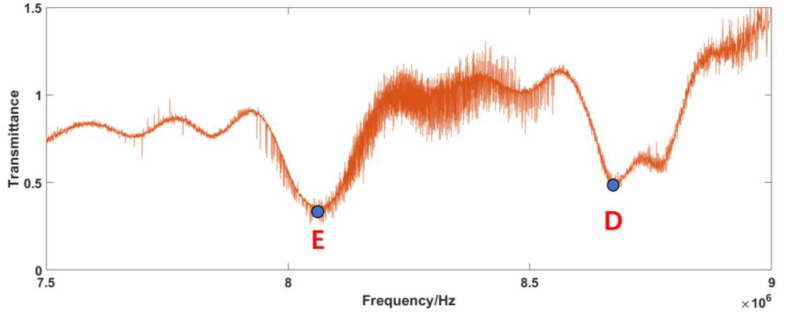
Normalized transmittance spectrum of the experiment signal.

**Figure 16 sensors-23-01103-f016:**
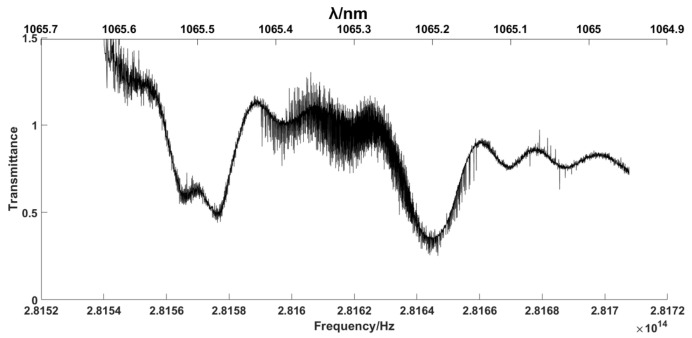
Experiment inversion from the RF domain to the OF domain.

**Figure 17 sensors-23-01103-f017:**
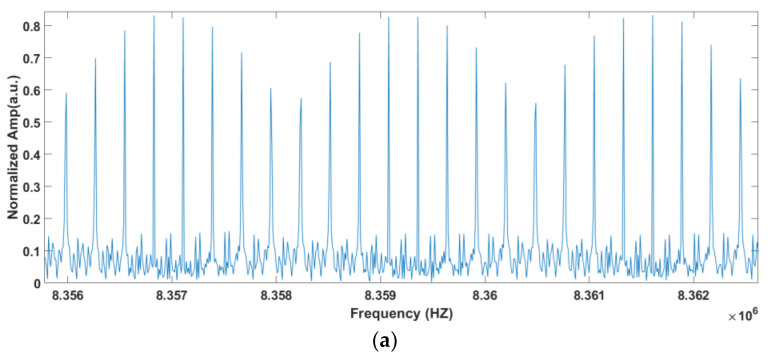
Comb component after the signal process in the experiment: (**a**): around the absorption peak, (**b**): around the peak of the whole spectrum.

**Figure 18 sensors-23-01103-f018:**
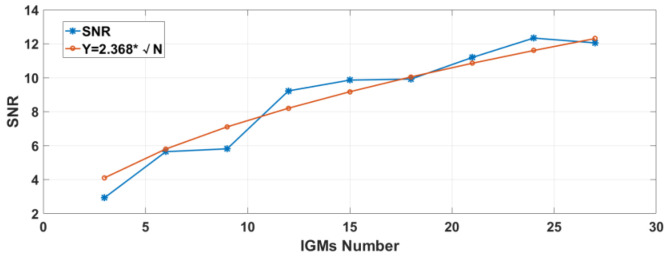
SNR varies with the sampling time.

**Table 1 sensors-23-01103-t001:** Parameters of the DCS system.

Devices	Parameters
Comb1 (SIG)	PRF: 26.64 MHz Output power: 71 mW Center wavelength: 1064.4 nm 3 dB width: 0.664 nm
Comb2 (LO)	PRF: 26.64 MHz Output power: 71 mW Center wavelength: 1064.4 nm 3 dB width: 0.714 nm
Δf_rep_	Approximately 281 Hz
LPF	3 dB 10.7 MHz
ADC sampling rate	100 M
Sampling time (simulation)	100 ms (28 IGMs)
FBG1	Center: 1064.15 nm BW = 0.035 nm (9.27 GHz) R = 73.6%
FBG2	Center: 1064.39 nm BW = 0.042 nm(11.13 GHz) R = 70.5%
Noise	$\delta T_{r}^{RF}(N)$: average jitter of 0.1 μs $\delta f_{c}^{RF}\left(N\right)$: average jitter of 0.2 Hz $\delta \varphi_{0}^{RF}(N)$: average jitter of 0.3 rad White noise: 0.15 (8.2 dB lower normalized) Pink noise: 0.025 (16 dB lower normalized) Quantization noise: 30 dB lower normalized

**Table 2 sensors-23-01103-t002:** Parameters used for the SNR calculation.

Parameters	Symbol	Value
Sequential detection series	*F*	1
Parallel detection series	*N_d_*	1
Spectral resolution	$v_{res}$	281 Hz
PRF	*f_r_*	26.640992826 MHz
Duty cycle	ε	1
Number of RF combs	*M*	14,223
Laser relative intensity noise	RIN	−130 dBc/Hz
Valid digits of ADC	*N*	10
Detector dynamic range	*D*	1774
RF comb spectral bandwidth	$\Delta v_{RF}$	Approximately 4 MHz
OFC spectral bandwidth	Δυ	Approximately370 GHz
Standard deviation of fast-change phase noise	$\sigma_{\varphi ,fast}^{}$	1.45 × 10^−5^ rad
Sampling time	*T*	100 ms

## Data Availability

The data presented in this study are available on request from the corresponding author.

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
