# Peer review of "An Investigation of All Fiber Free-Running Dual-Comb Spectroscopy"

_sensors, 2023, doi:10.3390/s23031103_

Round 1

Reviewer 3 Report

This manuscript is not well organized and written to some extents, such as title, figures and text. There are also some obvious errors in the main text. The authors should carefully prepare the paper before submission and let the whole manuscript satisfy the requirement of the journal. Especially, figures' quality needs to improve a lot. Some improvements or revisions suggestions are listed as follow:

1.     Figure 10, it is so hard to read this figure. Every part used in the experiment is not clear and messy, the reader can’t discriminate what it is. It is better to add a schematic of the set up to well show your system.

2.     Figures 12 and 13, the authors added inserts in Figs. 12(b and d), Figs. 13(b and d), but the corresponding scale are missing while they are not clean enough, the figure quality is also needed to improve.

3.     Page 9, line 288; page 12, line 380, frequency range should be “7-9 MHz”, “7.5-9 MHz” not “7MHz 9 MHz”, “7.5MHz 9 MHz”.

4.     Figure or Fig. should be used in the correct way.

5.     The title “Study of …” is not suitable for this paper, delete “Study of” or using the better words or phrases.

Round 2

Reviewer 1 Report

The authors have adequately addressed my comments. I recommend the paper to be considered for publication. 

The style of the refereneces are not unified in the current version. A preproduction review is needed.

Author Response

Dear Reviewer:

Thanks very much for your kind work and consideration on publication of our paper. On behalf of my co-authors, we would like to express our great appreciation to editor and reviewers.

The corrections in the paper and the responds to the reviewer’s comments are as flowing:

Q:The style of the references are not unified in the current version. A preproduction review is needed.

Answer:Thank you for the suggestions and recommendations, We are very sorry for our negligence of the style of the references. The style of the references has been adjusted and highlighted in the revised version.

Thank you and best regards

Authors of Sensor-2090456

Reviewer 2 Report

After revising all the answers of the authors and how they have managed to address most of my concerns, I recommend to publish the manuscript.

Author Response

Dear Reviewer:

Thanks very much for your kind work and consideration on publication of our paper. On behalf of my co-authors, we would like to express our great appreciation to editor and reviewers.

Thank you and best regards

Authors of Sensor-2090456

Reviewer 3 Report

The authors have addressed all comments from reviewers and revised the manuscript.

Author Response

(The authors gave the same response as above.)
